# Evaluating the Determinants of Substance Use in LGBTQIA+ Adolescents: A Scoping Review

**DOI:** 10.3390/ijerph21121579

**Published:** 2024-11-27

**Authors:** Eric Brown, Erini Abdelmassih, Fahad Hanna

**Affiliations:** 1Program of Public Health, Department of Health and Education, Torrens University Australia, Sydney, NSW 2010, Australia; eric.brown@health.torrens.edu.au; 2School of Pharmacy, University of Tasmania, Grosvenor St., Sandy Bay, Hobart, TAS 7001, Australia; erini.abdelmessih@utas.edu.au; 3Program of Public Health, Department of Health and Education, Torrens University Australia, Melbourne, VIC 3000, Australia; 4Higher Education College, Chisholm Institute, Melbourne, VIC 3000, Australia

**Keywords:** adolescent, drug abuse, illicit drug use, LGBTQ+, risk factors

## Abstract

**Background:** Research has consistently shown increased drug use among lesbian, gay, bisexual, transgender, intersex, queer/questioning, and asexual (LGBTQIA+) individuals. This is particularly the case among LGBTQIA+ adolescents. Substance use within this vulnerable community can propagate mental health issues, leading to psychiatric disorders, self-harm, and even suicide. Therefore, the objective of this scoping review was to evaluate the determinants of drug use among LGBTQIA+ adolescents. **Methods:** A comprehensive search of mainly primary research was conducted, using several databases. Peer-reviewed articles published between 2018 and 2023 were included. The scoping review was conducted using the framework outlined by Joanna Briggs Institute (JBI) and reported using the Preferred Reporting Items for Systematic Reviews and Meta-Analyses—Extension for Scoping Reviews (PRISMA-ScR) statement. **Results:** A total of 29 articles (including 400,194 participants) were included in the analysis. The articles reported that the main determinants of drug use among LGBTQIA+ adolescents include homelessness, peer–peer interactions, mental health, and protective factors. The articles reported that mental health issues, which were mainly triggered by rejection, were the main determinants of drug use among LGBTQIA+ adolescents. **Conclusions:** Findings from this scoping review provide relatively reliable evidence that homelessness, mental health, peer–peer interactions, and protective factors are the main determinants of illicit drug use among LGBTQ+ adolescents. Rigorous studies including large sample sizes and systematic reviews are needed to further confirm these findings and assist in developing interventions to combat the unusually high level of drug use among this group.

## 1. Introduction

Illicit drug use continues to be a worldwide public health issue, the negative effects of which are slowly expanding to every corner of the globe. If the expansion of drug use and abuse is an ordeal, then the development of new synthetic drugs is akin to Armageddon. Recent data from the United Nations Office on Drugs and Crime (UNODC) World Drug Report of 2022 reported record rises in illicit drug manufacturing, coupled with an increase in illicit drug use worldwide [1]. In the United States alone, there were over 100,000 illicit drug overdose deaths in 2021. This is a substantial increase in drug overdose-related deaths compared to the 50,000 deaths reported in 2015 [2]. As the number of illicit drug users increases worldwide, so do the social, human, health, and economic costs, ultimately manifesting as drug-related violence, increased crime rates, legal orders, rehabilitative services, and reduced work capacity/productivity [3].

There is an increased susceptibility to illicit drug use within certain populations in our society. Research has consistently shown increased illicit drug use in the following populations: young people, those with mental health conditions, homeless people, and lesbian, gay, bisexual, transgender, intersex, queer/questioning, and asexual (LGBTQIA+) people [4].

Individuals within the LGBTQIA+ community constantly face negative stigma and micro-aggressions in their day-to-day lives [5]. This enduring homophobia, bullying, discrimination, and peer/familial rejection results in the development of mental scars that ultimately manifest as mental illnesses. The ongoing stressors encountered by sexual-minority individuals form the basis of the Minority Stress theory that states that LGBTQIA+ individuals encounter various stressors that accumulate, producing poor mental health [6], which is further exacerbated by their internalised homophobia [7]. This is one possible explanation as to the mental health disparities seen in sexual and gender minorities when compared to their heterosexual counterparts [6]. To cope with mental health issues, stigma, or prejudice, many LGBTQIA+ people often turn to unhealthy coping mechanisms, so as to lessen the daily pressure exerted on their mental health and wellbeing [8]. One common mechanism explored by the LGBTQIA+ community is the use of illicit drugs. A recent study showed a significant association between identity (sexual orientation or gender)-related distress and drug use, especially among queer adolescents [9]. Compared to heterosexual adolescents, queer adolescents were approximately 190% more likely to consume illicit drugs [10]. During adolescence, the brain’s cognitive functions are still in the process of developing, which makes all young people more vulnerable to experimentation and potential addiction. This period of heightened susceptibility is not unique to any one group, and adolescents identifying as LGBTQ+ experience the same neurodevelopmental challenges as their peers [11]. 

Given the increasing trend in drug trades/trafficking, as well as the number of people belonging to sexual/gender minorities, particularly those of adolescent age, and given the significant negative consequences associated with illicit drug use, particularly among adolescents, this scoping review aimed to evaluate the determinants of drug use in LGBTQIA+ adolescents.

## 2. Research Design and Methods

### 2.1. Basic Framework

Analysing the literature is important in advancing health and clinical knowledge. The two main methods that may be used to analyse the literature are systematic reviews and scoping reviews, which differ in their objectives and methodologies. Systematic reviews are adopted to address significant clinical questions or provide evidence that may inform practice. On the other hand, scoping reviews tend to have wider scopes and are conducted to determine gaps in the existing knowledge and to clarify concepts. Therefore, the choice of approach may depend on the research aims and the depth of analysis needed [12,13]. Hence, to attain the aim above (the aim was to evaluate the determinants of drug use in LGBTQIA+ adolescents), a scoping review of the literature was conducted.

To achieve the aim of the scoping review, the “Joanna Briggs Institute (JBI) methodology for scoping review” was used [https://jbi.global/scoping-review-network/resources#, (accessed on 15 July 2024)]. Arksey and O’Malley first suggested this search framework in 2005 [14,15]. The scoping review was reported using the Preferred Reporting Items for Systematic Reviews and Meta-Analyses—Extension for Scoping Reviews (PRISMA-ScR) statement [13]. (The search strategy employed an iterative process and was guided by the research question. The guidelines established by the JBI for conducting a scoping review consist of the following steps: developing the research question(s)/topic, identifying the relevant literature, selecting the literature that meets the inclusion criteria, extracting results from the selected literature, and presenting the results [12] 

### 2.2. Developing the Research Question

Arksey and O’Malley suggest an iterative process to develop the research question. The following research question was developed to guide the present scoping review:

What are the determinants of drug use among LGBTQIA+ adolescents?

### 2.3. Identification of the Relevant Literature

A literature search was conducted in PubMed, ProQuest, and CINAHL to identify articles published between 2018 and 2023, using a combination of keywords and MESH terms for Adolescent, drug use, LGBTQ+, youth, and risk factors. Keywords including “Adolescent*, youth, drug abuse, drug use, LGBTQIA+, LGBT* and risk factors” were used, and the Boolean operators “OR”, “AND” were used as required. Multiple databases were chosen for this study to improve the results and reduce the risk of overlooking any eligible studies that could be used during our final appraisal. Furthermore, these specific databases were chosen as they provide extensive coverage of the health-related literature [16]. 

### 2.4. Study Selection

Article titles and abstracts were initially screened against the inclusion criteria to determine which articles would undergo full-text review. Then, the full text of the resulting articles was reviewed for inclusion. Also, the reference list of all included articles was searched for additional articles. Articles that were considered for inclusion included articles that provided original data (e.g., randomised controlled trials and observational studies), as well as systematic reviews; those that included participants who were both sexual minorities and adolescent populations (this is given the increasing trend in the number of adolescents belonging to sexual/gender minorities and given the significant negative consequences associated with illicit drug use among them); peer-reviewed articles; articles published in the English language; and those published between 2018 and 2023. We have limited the inclusion criteria to articles that were published within the last five years. This ensured that the analysis reflected the present status of drug use among the target cohort. Also, this time frame revealed drug use trends following the COVID-19 pandemic, which resulted in changes in drug consumption, with consequential mental effects occurring due to drug use, withdrawal, and overdose [17]. 

The exclusion criteria included articles that focused on the target cohort’s adult counterparts; opinion articles; those that did not align with our research question; and those that did not meet any of the inclusion criteria discussed above. 

Given that there are multiple definitions for the adolescent age group, we utilised the World Health Organization (WHO)’s definition of adolescents. According to the WHO, adolescents are defined as individuals within the second decade of life, that is, those who are 10–19 years of age [18]. 

### 2.5. Data Extraction

A qualitative descriptive approach was used to extract data. Articles retrieved from the PubMed, ProQuest, and CINAHL databases were compiled within the Mendeley citation manager, and duplicates were removed. The remaining articles were analysed, using their titles and abstracts for screening against the eligibility criteria. Following this, the articles underwent a full-text review, and those that were not relevant to the research question were eliminated from the final pool. Each included document was reviewed to ensure that it was relevant to the research question. A double-review process was undertaken during the two selection sub-steps: the title and abstract, and the full text. Disagreements regarding inclusion or exclusion were resolved by discussion. Relevant data, including each article’s aim, geographic location, and results were extracted and synthesised.

The PRISMA flow diagram was followed for study identification and to select the articles to be included in the final analysis.

### 2.6. Presenting/Reporting of Results

Once data extraction was completed, Braun and Clarke’s approach to thematic analysis was utilised to evaluate the [19]. This approach consists of six phases: 1. becoming familiar with the data; 2. producing initial codes for the data; 3. searching for potential themes; 4. reviewing themes; 5. defining and naming themes; and 6. reporting and analysing themes [19]. Phase 6, which involved reporting and analysing findings, was completed using the Preferred Reporting Items for Systematic Reviews and Meta-Analyses—Extension for Scoping Reviews (PRISMA-ScR) guidelines.

Findings were summarised using an iterative coding process. These were used to develop a series of categories that broadly captured the determinants of drug use among LGBTQIA+ adolescents.

## 3. Results

### 3.1. Search Results

A total of 10,081 articles were initially identified within the three Torrens University Australia databases (PubMed, EBSCO, and ProQuest). After removing duplicates (n = 114), 9967 articles underwent our screening process following the PRISMA chart used in systematic and scoping reviews, yielding a total of twenty-nine articles to be evaluated for this review (Figure 1).

### 3.2. Study Characteristics

All included articles were published between 2018 and 2023 and predominantly utilised quantitative methodologies. Eighteen articles were published in the United States of America (USA). The remaining studies were spread across a few countries: New Zealand, Australia, Thailand, South Africa, Canada, Singapore, and Brazil. Table 1 gives the detailed characteristics of the included studies.

A narrative account was prepared from the included studies, to evaluate the determinants of drug use in LGBTQIA+ adolescents. Notably, most articles examined multiple determinants simultaneously. Based on the research question, the extracted data were segregated thematically into four main themes (Figure 2). These are as follows:

#### 3.2.1. Homelessness

A homeless person is defined as a person who is currently living/sleeping in non-conventional accommodation, including staying on the street or in short-term/emergency accommodation, which can include living with friends/family [49]. Five studies explored the influence of homelessness on drug use among LGBTQ+ adolescents [22,23,28,34,36]. Studies were conducted in the USA, Canada, and New Zealand (NZ). The participant number varied from 77,559 participants to 14 participants. These studies showed that homelessness is a strong determinant of drug use among LGBTQ+ adolescents. One study showed that LGBTQ+ students (14–18 years old) who experienced homelessness were 196% more likely to use alcohol and 275% more likely to use hard drugs than their counterparts [22].Another article showed that adolescents with a history of foster care are more likely to use alcohol or illicit drugs, especially during sexual intercourse [34]. This study also showed that drug use/abuse was both a consequence of and a reason for homelessness [34]. Also, a study by Noble et al. explored the impacts of COVID-19 on adolescents experiencing homelessness [36]. The study showed that feelings of isolation/loneliness increased mental health instabilities and drug use among LGBTQ+ adolescents [36]. 

#### 3.2.2. Peer–Peer Interactions

For the purpose of this scoping review, we defined peer-to-peer interactions as interactions between LGBTQ+ adolescent individuals. These interactions can be non-verbal, verbal, or physical interactions. Also, these interactions can be positive (i.e., relationships) or negative (i.e., homophobia). Twelve studies explored the influence of peer–peer interactions on drug use among LGBTQ+ adolescents [21,22,23,24,29,31,32,35,39,42,44,47]. Studies were conducted in the USA, NZ, Thailand, South Africa, and Brazil. The participant number varied from 77,559 participants to 10 participants. The studies showed that peer–peer interaction is a determinant of drug use among LGBTQ+ adolescents. Researchers in Thailand explored social violence in Thai gender role conforming versus non-conforming students. The study showed that LGBTQ+ adolescent participants used substances that ranged from cannabis to injected drugs, mainly due to experiencing social violence [24]. Another study examined sexual violence and drug use in first-year female university students (sexual-minority vs. non-minority students). The study showed that sexual-minority female students were more likely to report illicit drug use than heterosexual women of the same age, mainly due to experiencing sexual violence [29]. Bisexual women within this study were also more likely to report new cases of sexual violence than their heterosexual counterparts [29]. 

#### 3.2.3. Mental Health

Mental health issues, including depression, internalised homophobia, social phobias, stigmas, and stress, were cited as determinants of illicit drug use in nine studies in this scoping review [22,23,27,31,34,35,37,38,43]. These studies reported that mental health is a strong determinant of drug use among LGBTQ+ adolescents. These studies were conducted in the USA, NZ, Australia, South Africa, and Singapore. The participant number varied from 77,559 participants to 14 participants. A USA-based study examined the relationship between peer victimisation, drug use, and depressive symptoms. The study reported that adolescents who reported drug use were 1.7 times more likely to endorse suicidal behaviour [31]. Similar findings were observed in a South African study that assessed substance misuse and depressive symptoms in gay/bisexual men [35]. In this study, participants who displayed high levels of internalised homophobia had higher odds of substance use and more depressive symptoms [35]. Another study showed that the delayed acceptance of sexual orientation or internalised homophobia were positively associated with smoking cigarettes and marijuana use [37]. 

#### 3.2.4. Protective Factors

The final theme observed within this scoping review relates to the protective factors of drug use among LGBTQ+ adolescents [21,26,32,39,46]. All five studies that cited this theme were conducted in the USA. The participant number varied from 77,559 to 10 participants. A systematic review that explored the beneficial effects of psychotherapeutic interventions in LGBTQ+ youths (14–22 years old) with a history of mental illness and drug abuse (Bochicchio et al., 2022) showed a significant reduction in drug use among participants that underwent psychotherapeutic interventions [21]. Cognitive reappraisal in this population had a negative association with illicit drug use [39]. Another study showed that resilience and gender-related pride have a negative association with gender-minority stressors among LGBTQ+ adolescents. This, in turn, lowers the future odds of drug use [32]. The study also showed that strong family function and social support groups are protective factors that may be associated with lower levels of gender-minority stress among LGBTQ+ adolescents [32]. However, a study by Whitton et al. that explored protective factors that may reduce the risk of substance use among LGBTQ+ adolescents reported that romantic involvement was associated with increased marijuana and other illicit drug use [46]. 

## 4. Discussion

This scoping review showed that homelessness, peer–peer interactions, and mental health are strong risk factors for drug use among LGBTQIA+ adolescents, with “protective factors” being negatively associated with drug use in this community.

Homelessness is a significant factor in drug use, particularly among young people. Currently, it is estimated that LGBTQIA+ people make up approximately 20–40% of the homeless population [28]. Many factors contribute to the cycle of homelessness within this population. A proximal cause of the increased homelessness in this population is drug use, which is both a cause of and a response to homelessness in this population [34]. Many studies suggested that the main reason for LGBTQIA+ adolescents becoming homeless was running away after being rejected by their families [50]. This parental rejection often results in ongoing mental health issues that linger into adulthood. These, in turn, lead to illicit drug use. A study by Ryan et al. examined lesbian, gay, and bisexual (LGB) adults’ mental health following parental rejection during their teenage years. The study reported that these individuals were 3.4 times more likely to use illegal substances than their adult counterparts [51]. Another study within this scoping review reported that strong family functioning and peer social support may have a protective effect against the use of illicit drugs [32].

Peer-to-peer interactions have been proven to have a significant impact on LGBTQIA+ adolescents’ drug use, including experimenting. Peers can greatly influence behaviours during this developmental stage. Research shows that adolescents often engage in substance use due to peer pressure, modelling behaviours or seeking acceptance within their social groups. These interactions shape attitudes toward substances, normalising or encouraging use among young people [52]. For example, one study within this scoping review reported that many transgender and lesbian individuals experience sexual violence [29]. It was also reported that bisexual women experience a significantly higher lifetime prevalence of physical violence and rape compared to heterosexual women [53]. These acts of violence result in the development of Rape Trauma Syndrome (RTS), which is characterised by three distinct phases that cumulate in victims feeling shame, guilt, anxiety, or depression [54]. To cope with these feelings, most victims adopt harmful practices, which may include suicidal ideation or drug use [55]. Another important peer-to-peer interaction in this group relates to sexual practices. Two studies in this scoping review explored the reasons why crystal methamphetamine tends to be used by LGBTQIA+ individuals during intercourse. Participants in these studies admitted to using crystal methamphetamine during sex to enhance sexual pleasure [30]. This concurs with a mainstream study by Whitton et al. that reported that romantic involvement may be associated with increased marijuana and other illicit drug use [46]. 

Our scoping review also identified that LGBTQIA+ adolescents tend to have numerous mental health problems, including depression, internalised homophobia, stigmas, and suicidal ideation. These issues are likely exacerbated by minority stress. Coping mechanisms during the adolescent years consists of three main strategies: support seeking, problem solving, and distraction/escape [56]. Given that adolescents often feel that they are alone and have no one to talk to, they frequently resort to distraction/escape coping mechanisms, which routinely end in experimentation with illicit drugs. This, unfortunately, worsens their mental health and, hence, there is a vicious cycle. This early onset of drug abuse (i.e., during adolescent years) often results in impaired development of the critical thinking skills and cognitive skills essential for successful adults [57]. Addressing these issues requires a dual approach that targets both positive and negative influences, empowering individuals within this community through interventions that promote adolescent health. These interventions should emphasise protective factors, foster resilience, and enhance quality of life by highlighting healthy adult strengths. Concurrently, schema therapy can be utilised to address underlying issues of abandonment and rejection, which are particularly relevant for this population [58].

This review had certain limitations. One of these limitations is that several of theincluded studies were cross-sectional, limiting the ability to establish causation between mental health and drug use. Nonetheless, the strength of the findings, supported by consistent results across multiple studies in this review, reinforces the observed association between mental health issues and substance use. Another limitation is related to the countries included in the review. Drug laws, such as what is legal and what is illegal, can vary significantly from one country to another and this may impact the behaviour and prevalence of drug use within any community.

## 5. Conclusions and Recommendations

This scoping review investigated the determinants of drug use among LGBTQIA+ adolescents. Homelessness, peer-to-peer interactions, and mental health factors, including positive (protective) mental health such as resilience, gender-related pride, and social support, were strong determinants of drug use among LGBTQIA+ adolescents. The results of this scoping review concur with those observed within adult LGBTQIA+ individuals. These disparities contribute to drug use in this community and among their adult counterparts, mainly exacerbated by minority stress. Increasing community resources to help address and improve coping strategies and increasing access to cognitive therapy for those with a history of trauma are paramount. Additionally, meaningful societal support and advocacy aimed at reducing stigma and discrimination through education and awareness may contribute to a reduction in drug use within this community. The findings from this scoping review may be instrumental for healthcare providers and policymakers in identifying strategies and designing tailored community-based programmes to reduce and break the cycle of drug use in this vulnerable population. Further research including systematic reviews and meta-analyses is needed to confirm the above findings.

## Figures and Tables

**Figure 1 ijerph-21-01579-f001:**
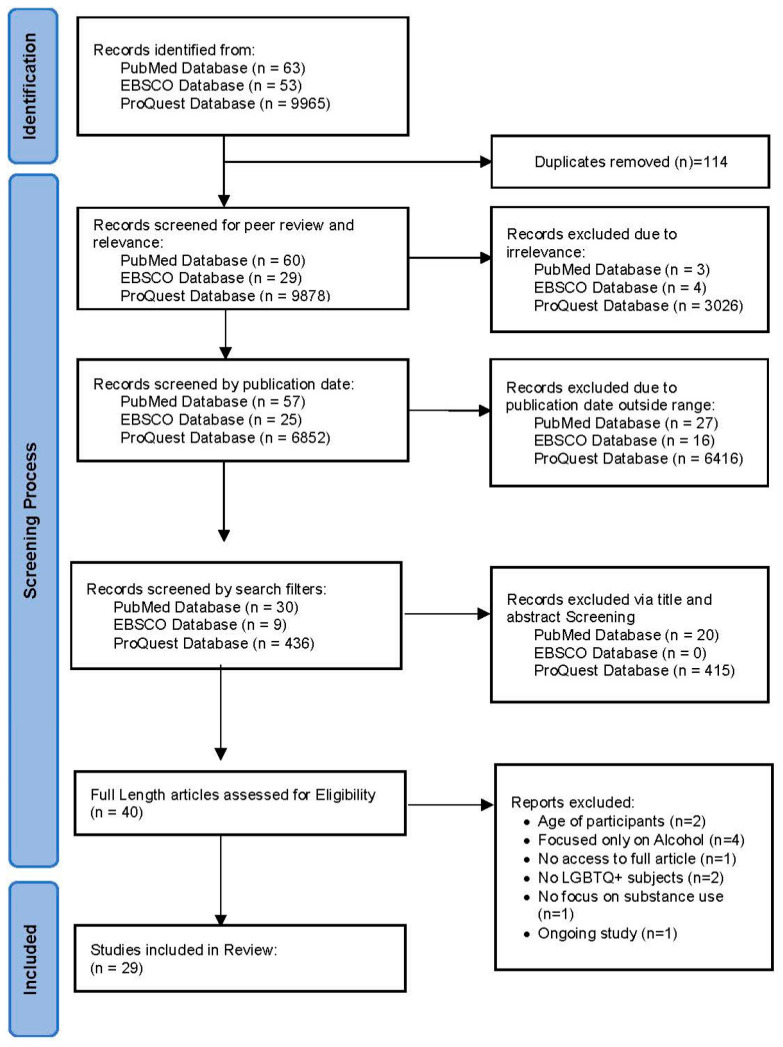
Review PRISMA flowchart.

**Figure 2 ijerph-21-01579-f002:**
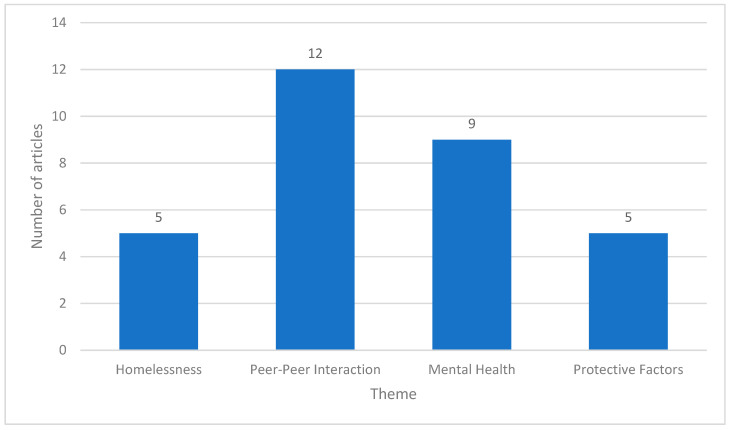
General themes that emerged from the reviewed studies.

**Table 1 ijerph-21-01579-t001:** Characteristics of the included studies.

#	Author Year and Country	Article Title	Population (n) Age Range	Aims/Parameters Examined	Study Design/Key Findings
**1**	Arrington-Sanders et al., 2020. USA [20].	Providing unique support for health study among Young Black and Latinz Men who have sex with Men and Young Black and Latinx Transgender Women living in 3 urban Cities in the USA: Protocol for a Coach-Based Mobile-Enhanced Randomized Control Trial.	Young Black and Latinx men who have sex with men, and transgender women. Aged 15–24 years. N = 402 participants.	To assess mobile-enhanced interventions compared to standard care, to increase engagement and retention in HIV, PrEP, and substance use treatment care.	- Most participants (median age, 21.6) reported a history of substance use: alcohol (77.6%) and other drugs (opioids, amphetamine, cocaine) (21.9%). - A total of 24.1% reported having housing issues. - A quarter reported engaging in transactional sex for housing, money, or food.
**2**	Bochicchio et al., 2022. USA [21].	Psychotherapeutic interventions for LGBTQ+ youth: a systematic review.	LGBTQ+ adolescents. Age range: 14–22 years old (yo). (n) = 10 articles (n = 822) that ranged from 10 to 268 participants.	Psychotherapeutic interventions for LGBT+ adolescents with mental illness and substance abuse.	- The results showed a significant reduction in mental health symptoms and drug use for all youth in the study.
**3**	Cutuli et al., 2020. USA [22].	Adolescent Homelessness and Associated Features: Prevalence and Risk Across Eight States.	LGBTQ+ youth from 14 to 18 yo in specific states in the USA. N = 77,559.	To test for positive associations between homelessness and key indicators.	- The results revealed that students who experienced homelessness were 196% more likely to use alcohol and 275% more likely to use hard drugs. - LGB youths were 143% more likely to experience homelessness.
**4**	Damian et al., 2022. USA [23].	Understanding the Health and Health-Related Social Needs of Youth Experiencing Homelessness: A photovoice Study.	LGBTQ+ youths (14–24 yo) experiencing homelessness during the COVID-19 pandemic in Connecticut. N = 14.	Record the everyday reality for homeless youth.	Participants described a vicious cycle of alcohol and substance use that made housing insecurity worse.
**5**	Do et al., 2020. Thailand [24].	Social Violence Among Thai gender role conforming and non-conforming secondary school students: Types, prevalence, and correlates.	Secondary school students who are trans or same-sex-attracted. Age, 13–20 yo. N = 2070.		Substance use was found in 98 (4.8%) participants and included cannabis, amphetamine pills, crystal methamphetamine, ecstasy/MDMA, sleeping pills, and injected-drug use.
**6**	Eastwood et al., 2021. USA [25].	Young Transgender Women of Colour: Homelessness, Poverty, Childhood Sexual Abuse and Implications for HIV Care.	HIV+ young (18–24 yo) transgender women of colour. N = 102.	To engage and retain transgender women of colour with HIV care to manage viral load suppression and determine factors contributing to sustained healthcare.	A total of 15.6% of participants reported drug dependence.
**7**	Eisenberg et al., 2020. USA [26].	Supportive Community Resources Are Associated with Lower Risk of Substance Use among Lesbian, Gay, Bisexual, and Questioning Adolescents in Minnesota.	LGBQ high school students. N = 2454.	To examine stigma and support and their association with substance use in LGBQ youth.	- Marijuana use: 8.5–14.6%. - Prescription drug misuse: 6.1–14%. - Other drug use total: 13.5%.
**8**	Filia et al., 2022. AUS/NZ [27].	Social inclusion, intersectionality, and profiles of vulnerable groups of young people seeking mental health support.	Young people 12–25 yo attending a Headspace Centre for mental health- ± substance use-related issues. Age range, 12–25 yo. N = 1107.	To examine social inclusion across the specific domains of housing, employment, study, and alcohol and other drugs.	- A total of 17% reported substance use. - A total of 70% reported substance use in the past 3 mos. - A higher usage of different substances was seen in the 18–25 yo group. - In total, 18% admitted that their substance use had resulted in them failing to complete obligations.
**9**	Fraser et al., 2019. NZ [28].	LGBTIQ+ Homelessness: A Review of Literature.	“LGBT Homelessness”, “Queer Homelessness”, and “LGBT Housing First”—essentially, LGBTIQ+ homelessness. N = 53 articles.	To examine the intersecting factors associated with homelessness and LGBTIQ+ homelessness.	- Substance use is a proximal cause of homelessness. - LGBTIQ+ homeless people have higher rates of substance use. - Transgender homeless people have even higher rates. - Homeless LGBTIQ+ people were more likely to report the usage of 20 out of 21 illicit substances compared to homeless non-LGBTIQ people.
**10**	Griffin et al., 2022. USA [29].	Sexual Violence and Substance Use among First-Year University Women: Differences by Sexual Minority Status.	First-year female university students from 14 USA universities. N = 974. Mean age, 19.1.	To examine the rates of sexual violence, perpetration, and substance use seen in female first-year university students.	- Sexual-minority women reported more frequent cigarette smoking, marijuana use, intoxication, use of club drugs, and overall illicit drug use. - Sexual-minority women reported illicit drug use at a greater proportion vs. heterosexual woman. - Bisexual women and those who consumed illicit drugs were more likely to report new cases of sexual violence.
**11**	Hammoud et al., 2020. AUS [30].	Biomedical HIV Protection Among Gay and Bisexual Men Who Use Crystal Methamphetamine.	Gay/bisexual men >16 yo who had had sex with another man in the last 12 months and lived in Australia. Median age, 35 (16–81 yo). N = 1367.	To investigate the relationship between crystal methamphetamine use and HIV risk behaviours in relation to biomedical prevention.	- A total of 13% reported previous crystal methamphetamine use but NOT in the last 6 months. - A total of 40.2% reported using crystal methamphetamine monthly or more often. - Of those that admitting to using drugs in the previous 6 months, 85.2% reported using them to enhance sexual pleasure, engage in chemsex (63.1%), and have “better sex” (67.5%).
**12**	Hatchel et al., 2019. USA [31].	Predictors of Suicidal Ideation and Attempts among LGBTQ Adolescents: The Roles of Help-seeking Beliefs, Peer Victimization, Depressive Symptoms, and Drug Use.	LGBTQ+ high school students participated in a randomised clinical trial testing the effects and sources of strengths. Mean age, 15 yo. N = 713 (LGBTQ).	To examine whether peer victimisation, drug use, depressive symptoms, and help-seeking beliefs predict suicidal ideation/attempts among LGBTQ adolescents.	- Adolescents are 1.7 times more likely to report suicidal behaviour if they report drug use.
**13**	Ksatz-Wise et al., 2021. USA [32].	Longitudinal effects of gender minority stressors on substance use and related risk and protective factors among gender minority adolescents.	Gender-minority adolescents in the US from the Trans Teen and Family Narratives (TTFN) project. Aged 13–17 yo. N = 30.	To determine the effects of minority stressors (gender) on substance use among gender-minority adolescents and the related risk/protective factors.	- Gender-minority adolescents *appear* to use substances to help cope with gender-minority stressors. - Wave 1: 17% of participants reported the use of one substance, and 4% reported using multiple substances. - The average age at first use was 13.7 for tobacco, 13.4 for alcohol, and 14.4 for marijuana. - Wave 5 (2 years after Wave 1): 56% reported the use of one substance, and 32% reported multiple substance use. - Internalised transphobia had a significant effect on substance use risk, except for tobacco. - Depressive/anxious symptoms did not show a significant effect on substance use. - Resilience and gender-related pride had a negative association with gender-minority stressors, which in turn lowered the odds of future substance use. - Family functioning and peer social support had protective effects against alcohol at lower levels of gender-minority stress.
**14**	Költö et al., 2019. Europe [33].	Romantic Attraction and Substance Use in 15-Year-Old Adolescents from Eight European Countries.	Same- and both-gender-attracted 15 yo adolescents from different European countries. Average age, 15.55. N = 14,545.	To explore the association between being of a sexual minority and different substance use behaviours.	- Respondents that reported having been in love with people of multiple genders had the highest prevalence of substance use (cigarettes, 33.6%; alcohol, 51.2%; cannabis, 20.6%). - Never having been in love was associated with lower odds of substance use for boys and girls. - Gay/bisexual adolescents had higher odds of using two or all three substances compared to heterosexual young people. - Gay adolescent males had significantly higher odds of multiple substance use.
**15**	Maria et al., 2020. USA [34].	Sexual risk classes among youth experiencing homelessness: Relation to childhood adversities, current mental symptoms, substance use, and HIV testing.	Youths between the ages of 13 and 24 yo who were homeless or had unstable housing. Age, 13–24 yo. N = 416 (final analysis).	To determine whether different subgroups of youth with different types of sexual risk behaviours experience homelessness and to examine the associations between potential classes and other variables.	- The high-risk class (defined by researchers) reported significantly higher levels of synthetic marijuana use, alcohol use, mental health diagnoses, and frequent HIV testing. - Youths that were not sexually active had the lowest rates of marijuana use, alcohol use, and HIV testing. - Youths with any history of foster care were more likely to be diagnosed with HIV/STI and to use alcohol or drugs during sex. - Substance use among youths experiencing homelessness is both a cause and a consequence of living on the streets.
**16**	Metheny et al., 2022. South Africa [35].	Correlates of Substance Misuse, Transactional Sex, and Depressive Symptomatology Among Partnered Gay, Bisexual and Other Men Who Have Sex with Men in South Africa and Namibia.	Gay, bisexual, and other men who have sex with men in South Africa and Nambia. Age, 18–24. N = 152.	To assess the association between three major HIV risk factors in gay/bisexual men in Southern Africa.	- High rates of substance misuse were seen in this population, with 77.5% (n = 341) reporting problematic alcohol use and 52.7% (n = 232) reporting non-prescription drug use. - Higher education was associated with lower odds of engaging in risky alcohol or drug use. - Being arrested was significantly associated with higher odds of recent substance misuse. - Higher levels of internalised homonegativity were associated with higher odds of substance misuse and more depressive symptoms. - Experience of discrimination was significantly associated with substance misuse, transactional sex, and depressive symptoms.
**17**	Noble A et al., 2022. Canada [36].	“I feel like I’m in a revolving door, and COVID has made it spin a lot faster”: The impact of the COVID-19 pandemic on youth experiencing homelessness in Toronto, Canada.	Youths experiencing homelessness in Toronto, Ontario, Canada, with a focus on particular sub-groups, mainly 2SLGBTQ, Black youths, and newcomer youths. Age, 16–24. N = 45 youths. N = 31 staff members. N = 9 2SLGBTQ (20%).	To appraise the impact of the COVID-19 pandemic on youths experiencing homelessness.	- Staff: the structural changes caused by the pandemic (changes in shelter sector and increased barriers to obtaining housing or employment) resulted in youths feeling increased feelings of isolation/loneliness and challenges with mental health and substance use.
**18**	Ong et al., 2021. Singapore [37].	Association between sexual orientation acceptance and suicidal ideation, substance use, and internalised homophobia amongst the pink carpet Y cohort study of young gay, bisexual, and queer men in Singapore.	Gay/bi/questioning men living in Singapore who are either HIV-negative or unsure about their HIV status. Age, 18–25 yo (mean, 21.9). N = 564.	To explore the associations between delayed acceptance of sexual orientation and health-specific outcomes relating to gay/bi/questioning men in Singapore.	- Delayed acceptance of orientation was positively associated with smoking cigarettes and marijuana usage.
**19**	Pike et al., 2023. USA [38].	A scoping review of survey research with gender minority adolescents and youth in low and middle-income countries.	Peer-reviewed articles published in English that utilise surveying data to explore gender-minority youth experiences. N = 33 articles analysed.	To explore the different ways in which the experiences of gender-minority youths are studied.	- Discrimination and instability at home were the main drivers of substance use. - Social victimisation due to sexual orientation or gender identity was associated with a higher likelihood of drug use.
**20**	Scheer J et al., 2021. USA [39].	Intimate Partner Violence and Illicit Substance Use Among Sexual and Gender Minority Youth: The Protective Role of Cognitive Reappraisal.	Self-identified sexual- and gender-minority youths between the ages of 18 and 25. N = 149. Age, 18–25 yo.	To examine cognitive reappraisal as a moderator in the various forms of intimate partner violence and illicit substance use among sexual- and gender-minority youths.	- In the past 6 months, participants reported the use of cocaine (24.8%), stimulants (22.8%), heroin (20.8%), and hallucinogens (24.8%). - Physical abuse was associated with illicit substance use. - Cognitive reappraisal was negatively associated with illicit substance use. - Young racial-/ethnic-minority participants were more likely to report illicit substance use in the past 6 months. - Interaction between identity abuse and cognitive reappraisal was significantly associated with less illicit substance use, and the same was true for those experiencing physical abuse. - Psychological abuse and cognitive reappraisal were not significantly associated with illicit substance use.
**21**	Schuler et al., 2019. USA [40].	Differences in substance use disparities across age groups in a national cross-sectional survey of Lesbian, Gay, and Bisexual Adults.	LGB adults of different age groups (18–25, 26–34, 35–49). N = 76,354, LGBTQ+ = 4868.	To examine LGB disparities and recent substance use in different age groups and compare these to those of their heterosexual counterparts.	- Lifetime marijuana use was significantly higher for gay men 18–25 yo (65%). - Past-year marijuana use was significantly elevated in gay men 18–25 yo (46%). - Past-year marijuana use rates were significantly higher among bisexual women of all age groups and among L/G women 18–25 yo (52%). - Lifetime use of hallucinogens, cocaine, and inhalants was significantly elevated in gay men. - Past-year illicit drug use was significantly elevated in gay men across all age groups. - Factors associated with the sustained use of illicit drugs into middle adulthood among gay and bisexual men include attendance at LGBT clubs or circuit parties, as well as sexualised drug use.
**22**	Schuler et al., 2020. USA [41].	Substance Use Disparities at the Intersection of Sexual Identity and Race/Ethnicity: Results from the 2015–2018 National Survey on Drug Use and Health.	LGB adults of different ages groups and races/ethnicities from the 2015–2016 NSDUH. (n) = 168,560; LGB = 11,389.	To examine the differences in the presence and magnitude of substance use disparities in LGB adults across different races/ethnicities.	- Disparities were constituently greater in magnitude for Black and Hispanic LGB women compared with White LGB women. - Few disparities were observed among men; the magnitude of observed disparities did not differ by race/ethnicity.
**23**	Seekaew et al., 2019. Thailand [42].	Sexual patterns and practices among men who have sex with men and transgender women in Thailand: A qualitative assessment.	Thai men who have sex with men and transgender women, living in Bangkok, Thailand, over 18 yo. N = 12 MSM. N = 13 TGW. Median age: MSM = 33.1 (29.9–35.7), TGW = 25.8 (23.4–29.1).	To understand the diversity of men who have sex with men and transgender women in Thailand and to identify sexual patterns and themes in men who have sex with men in Bangkok.	- Many participants turned to alcohol and drugs (cannabis, poppers, amphetamines, and cocaine) to enhance their sexual drive, performance, and experience.
**24**	Sharma et al., 2019. USA [43].	Variations in Testing for HIV and Other Sexually Transmitted Infections Across Gender Identity Among Transgender Youth.	Individuals between 15 and 24 yo currently living in the USA that identify as non-cisgender and have never been diagnosed with HIV but were willing to conduct a rapid home HIV test. N = 186. Age range, 15–24 yo.	To quantify HIV and other STI testing levels and examine the variations in testing levels across three categories of gender identity: transgender men, transgender women, and nonbinary individuals.	In total, 71.5% of participants admitted that they use drugs.
**25**	Soares et al., 2023. Brazil [44].	Important steps for PrEP uptake among adolescent men who have sex with men and transgender women in Brazil.	Participants were between the ages of 15 and 19 yo and lived within the testing city. Age, 15–19. N = 751.	The aim of this study was to analyse the factors associated with drug use among adolescent men who have sex with men and transgender women in Brazil.	In total, 31.5% of participants reported drinking alcohol and 32.5% reported using drugs.
**26**	Watson R et al., 2020. USA [45].	Substance Use among a National Sample of Sexual and Gender Minority Adolescents: Intersections of Sex Assigned at Birth and Gender Identity.	Participants aged between 13 and 17 yo, who self-identified as being in a sexual or gender minority and resided in the US. N = 11,129. Age range, 15–17 yo.	The aim of this study was to test whether current gender identity and sex at birth were key factors in substance use among a large sample.	- Transgender adolescents had higher prevalence rates of marijuana/cigarette use compared to cisgender and nonbinary/genderqueer adolescents. - Gender identity was significantly associated with increased odds of substance use.
**27**	Whitton et al., 2018. USA [46].	Effects of romantic involvement on substance use among young sexual and gender minorities.	Gender-minority youths living in Chicago. N = 248. Age range, 16–20 yo.	The aim of this study was to identify protective factors that reduced the risk of substance use among sexual- and gender-minority adolescents.	- Romantic involvement was associated with less drinking in all participants, but participants also reported more cigarette smoking (26%). - Bisexuals had increased marijuana and other illicit drug use. - Participants reported smoking 26% more cigarettes when romantically involved. - Bisexuals reported increased marijuana use (rate ratio, 2.31) and other illicit drug use (odds ratio = 2.39).
**28**	Wichaidit et al., 2021. Thailand [47].	Disparities in behavioural health and experiences of violence between cisgender and transgender Thai adolescents.	Data from The National School Survey on Alcohol Consumption, Substance Use and Other Health-Risk Behaviours (a cross-sectional survey). The participants were stratified based on sex assigned at birth and their gender identity. N = 31,898.	The objective of this study was to assess the extent of behavioural health outcomes and violence among respondents of the National School Survey on Alcohol Consumption, Substance Use and Other Health-Risk Behaviours.	- Transgender boys had a higher prevalence of lifetime history of using illicit drugs compared to cisgender girls, particularly in the use of methamphetamine pills.
**29**	Yockey R and Barnett T, 2023. USA [48].	Past-Year Blunt Smoking among Youth: Differences by LGBT and Non-LGBT Identity.	LGBT+ youths. Age range, 14–17 yo. N = 7518.	The aim of this study was to investigate the past year’s marijuana and tobacco use among a national sample of adolescents and compare the difference between LGBT+ youths and non-LGBT youths.	LGBTQ+ youth were 2.17 times (95% CI 1.86, 2.54) more likely to report drug use than non-LGBT+ youth.

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
