# Peer review of "Evaluating the Determinants of Substance Use in LGBTQIA+ Adolescents: A Scoping Review"

_ijerph, 2024, doi:10.3390/ijerph21121579_

Round 1
Reviewer 1 Report
Comments and Suggestions for Authors
The introduction should be revised to provide a more precise and evidence-based opening for the academic paper. The broad statement about the worldwide public health issue of illicit drug use, including the overly dramatic comparison to Armageddon, should be replaced with a factual statement that clearly outlines the scope and significance of the issue. The various statistics provided in the introduction should be more explicitly connected to the overall argument or purpose of the paper. For example, when discussing the increase in drug-related deaths due to synthetic opioids like fentanyl, it should be explained why this is particularly relevant to the focus on LGBTQIA+ adolescents. To improve readability, the introduction should be reorganized to create a clearer narrative flow. It is suggested to start by establishing the global context, then focus on at-risk populations, and finally introduce the theoretical framework that guides the study. Subjective phrases such as "dramatic increase" and "nightmare" should be replaced with more precise academic language. The term "susceptibility to illicit drug use" should be clarified to avoid ambiguity by specifying whether it refers to a higher likelihood of initial use, addiction, or both.
The research design and methods section of the paper requires improvement. It mentions that a scoping review was chosen as the appropriate method, it lacks a detailed rationale for this decision. It is important to expand on why the scoping review methodology, specifically the Joanna Briggs Institute (JBI) approach, was most suitable for addressing the research question and how it adds value compared to other review types such as systematic reviews or meta-analyses. The section briefly lists the databases and keywords used for the search, but it would benefit from a more detailed explanation of the search strategy. It is important to include the rationale behind selecting specific databases and describe the process of combining keywords and MESH terms. Any limitations in the search strategy or database selection that could impact the comprehensiveness of the literature review should be mentioned. The criteria for including and excluding articles are briefly mentioned, but the rationale behind these choices requires further elaboration. It is important to explain why the study period was limited to 2018-2023 and how this timeframe ensures the relevance of the findings. Also, clarify why certain types of studies, such as those involving adult populations, were excluded and how this decision aligns with the study's objectives. The section mentions that Braun and Clarke's approach to thematic analysis was used, but it lacks detail on how the data was extracted and managed before analysis. It is crucial to outline the specific steps taken to ensure data consistency and reliability during extraction and coding. Provides more context on how thematic analysis was applied to identify determinants of drug use among LGBTQIA+ adolescents. Although the six phases of Braun and Clarke's thematic analysis are listed, the description does not explain how these phases were implemented in the context of the study. Including more specific examples or details about how themes were generated, reviewed, and refined would be beneficial. Discussing any challenges encountered during the analysis and how they were addressed would strengthen the methods section. The methods section does not address ethical considerations related to the scoping review. Even though scoping reviews typically do not involve primary data collection, it is important to mention how ethical guidelines were followed, particularly in the context of sensitive topics like drug use among LGBTQIA+ adolescents.
The results section needs improvement. Outline the criteria used for screening articles, how duplicates were identified, and the rationale for selecting the final 29 articles. This will enhance transparency and allow readers to better understand the systematic approach taken. Provide more detailed information about the types of quantitative methodologies employed and any significant variations in study design or focus. Highlighting these aspects will provide readers with a deeper understanding of the scope and quality of the included research. The thematic breakdown into four main themes is a good approach; however, the depth and consistency of analysis across these themes could be improved. Consider providing more specific examples or findings from the included studies to substantiate the narrative for each theme. Ensure each theme is discussed with a similar level of detail to create a more balanced and comprehensive section. Check that Figure 1 (PRISMA flowchart) and Figure 2 (general themes) are clear, detailed, and effectively labeled. Figure 1 should clearly show the numbers and reasons for exclusion at each stage of the review process. Figure 2 could be more visually descriptive, perhaps by including sub-themes or key findings under each main theme. In section 3.2.1 on homelessness, the studies presented show a strong correlation between homelessness and drug use among LGBTQ+ adolescents. Provide a more in-depth discussion on how each study defines and measures homelessness and its impact to enhance the analysis. Similarly, for sections on peer-to-peer interactions, mental health, and protective factors, expand on how these determinants were operationalized and measured across different studies.
The discussion in this manuscript currently mixes correlations and causations when discussing homelessness, drug use, and mental health among LGBTQIA+ adolescents. It would be helpful to clearly distinguish between factors that are correlational and those that have stronger evidence of causality. This distinction would allow readers to have a clearer understanding of the complexity of these issues. The discussion outlines several determinants of drug use, it lacks depth in exploring the underlying mechanisms that connect these factors. Providing more elaboration on how factors like peer interactions and mental health directly influence drug use would strengthen the discussion. Incorporating potential interventions or strategies based on the findings would provide valuable insights for practitioners and policymakers. It is important for the discussion to acknowledge the limitations of the studies reviewed, including any gaps in the literature, such as underrepresented groups within the LGBTQIA+ community or geographic biases in the cited studies. This acknowledgement would add rigour and transparency to the review. To improve the structure of the discussion, related findings should be grouped more cohesively. For example, consolidating all points about mental health into a single coherent paragraph instead of dispersing them throughout would enhance readability and comprehension. Providing a broader context about the social and environmental factors that contribute to the vulnerability of the LGBTQIA+ population to homelessness and drug use would be beneficial. This contextual background would help readers better understand the systemic nature of these issues.
Comments on the Quality of English Language
Minor editing of English language required.
Author Response
Responses to Reviewer 1:
Comments and Suggestions for Authors
The introduction should be revised to provide a more precise and evidence-based opening for the academic paper. The broad statement about the worldwide public health issue of illicit drug use, including the overly dramatic comparison to Armageddon, should be replaced with a factual statement that clearly outlines the scope and significance of the issue. The various statistics provided in the introduction should be more explicitly connected to the overall argument or purpose of the paper. For example, when discussing the increase in drug-related deaths due to synthetic opioids like fentanyl, it should be explained why this is particularly relevant to the focus on LGBTQIA+ adolescents. To improve readability, the introduction should be reorganized to create a clearer narrative flow. It is suggested to start by establishing the global context, then focus on at-risk populations, and finally introduce the theoretical framework that guides the study. Subjective phrases such as "dramatic increase" and "nightmare" should be replaced with more precise academic language. The term "susceptibility to illicit drug use" should be clarified to avoid ambiguity by specifying whether it refers to a higher likelihood of initial use, addiction, or both.
Response: The authors thank the reviewer for the constructive comment re the content and flow of the introduction and have adjusted the text accordingly. Please see the highlighted parts. Please also note that some of the drug use content in the introduction may be related to adolescents in general and with that, we highlight that it may impact those from LGBTQIA+ in a way that is amplified due to other factors.
Also, some wording and expressions were replaced following the reviewer’s suggestions- see highlighted words in the manuscript
The research design and methods section of the paper requires improvement. It mentions that a scoping review was chosen as the appropriate method, it lacks a detailed rationale for this decision. It is important to expand on why the scoping review methodology, specifically the Joanna Briggs Institute (JBI) approach, was most suitable for addressing the research question and how it adds value compared to other review types such as systematic reviews or meta-analyses. The section briefly lists the databases and keywords used for the search, but it would benefit from a more detailed explanation of the search strategy. It is important to include the rationale behind selecting specific databases and describe the process of combining keywords and MESH terms. Any limitations in the search strategy or database selection that could impact the comprehensiveness of the literature review should be mentioned. The criteria for including and excluding articles are briefly mentioned, but the rationale behind these choices requires further elaboration. It is important to explain why the study period was limited to 2018-2023 and how this timeframe ensures the relevance of the findings. Also, clarify why certain types of studies, such as those involving adult populations, were excluded and how this decision aligns with the study's objectives. The section mentions that Braun and Clarke's approach to thematic analysis was used, but it lacks detail on how the data was extracted and managed before analysis. It is crucial to outline the specific steps taken to ensure data consistency and reliability during extraction and coding. Provides more context on how thematic analysis was applied to identify determinants of drug use among LGBTQIA+ adolescents. Although the six phases of Braun and Clarke's thematic analysis are listed, the description does not explain how these phases were implemented in the context of the study. Including more specific examples or details about how themes were generated, reviewed, and refined would be beneficial. Discussing any challenges encountered during the analysis and how they were addressed would strengthen the methods section. The methods section does not address ethical considerations related to the scoping review. Even though scoping reviews typically do not involve primary data collection, it is important to mention how ethical guidelines were followed, particularly in the context of sensitive topics like drug use among LGBTQIA+ adolescents.
Response: The authors thank the reviewer again for the detailed comment and have now adjusted the methodology section to reflect their suggestions. See highlighted sections – the entire methodology section was rewritten in this instance
The results section needs improvement. Outline the criteria used for screening articles, how duplicates were identified, and the rationale for selecting the final 29 articles. This will enhance transparency and allow readers to better understand the systematic approach taken. Provide more detailed information about the types of quantitative methodologies employed and any significant variations in study design or focus. Highlighting these aspects will provide readers with a deeper understanding of the scope and quality of the included research. The thematic breakdown into four main themes is a good approach; however, the depth and consistency of analysis across these themes could be improved. Consider providing more specific examples or findings from the included studies to substantiate the narrative for each theme. Ensure each theme is discussed with a similar level of detail to create a more balanced and comprehensive section. Check that Figure 1 (PRISMA flowchart) and Figure 2 (general themes) are clear, detailed, and effectively labeled. Figure 1 should clearly show the numbers and reasons for exclusion at each stage of the review process. Figure 2 could be more visually descriptive, perhaps by including sub-themes or key findings under each main theme. In section 3.2.1 on homelessness, the studies presented show a strong correlation between homelessness and drug use among LGBTQ+ adolescents. Provide a more in-depth discussion on how each study defines and measures homelessness and its impact to enhance the analysis. Similarly, for sections on peer-to-peer interactions, mental health, and protective factors, expand on how these determinants were operationalized and measured across different studies.
Response: Please see modifications of the figure and other related parts in response to this comment. In relation to subthemes this is something we can adopt in future work. Thank you for the useful input
The discussion in this manuscript currently mixes correlations and causations when discussing homelessness, drug use, and mental health among LGBTQIA+ adolescents. It would be helpful to clearly distinguish between factors that are correlational and those that have stronger evidence of causality. This distinction would allow readers to have a clearer understanding of the complexity of these issues. The discussion outlines several determinants of drug use, it lacks depth in exploring the underlying mechanisms that connect these factors. Providing more elaboration on how factors like peer interactions and mental health directly influence drug use would strengthen the discussion. Incorporating potential interventions or strategies based on the findings would provide valuable insights for practitioners and policymakers. It is important for the discussion to acknowledge the limitations of the studies reviewed, including any gaps in the literature, such as underrepresented groups within the LGBTQIA+ community or geographic biases in the cited studies. This acknowledgement would add rigour and transparency to the review. To improve the structure of the discussion, related findings should be grouped more cohesively. For example, consolidating all points about mental health into a single coherent paragraph instead of dispersing them throughout would enhance readability and comprehension. Providing a broader context about the social and environmental factors that contribute to the vulnerability of the LGBTQIA+ population to homelessness and drug use would be beneficial. This contextual background would help readers better understand the systemic nature of these issues.
Response: Great suggestion. The discussion has been revised accordingly. See the updated section with highlights
Thanks again for the input that has undoubtedly added value to the manuscript
Reviewer 2 Report
Comments and Suggestions for Authors
I was very happy to read this scoping review. The method is appropriate when one wants to attempt an initial exploration within a still poorly reviewed literature. Important thoughts and insights for future research and clinical and prevention practice emerge from the authors' work. The introduction is pertinent (perhaps I would reduce the part on drug emergence, which is an established fact) and also include the concept of internalized homophobia as a factor that impacts the psychological health of young people (https://doi.org/10.1080/19419899.2018.1476905) and could be the consequence of rejection by peers, peers and the community at large.
The authors argued the criteria for inclusion and exclusion well and also explained why they focused only on such a narrow time frame. Usually in scoping reviews, gray literature is also used, but because scoping reviews focused only on peer-reviwed articles are flourishing, I ask the authors to specify this fact.
I would avoid stating for each specific factor the sample size of the included research. In addition, I would prepare a table summarizing the factors identified and the studies that identified them, perhaps inserting with an X when a particular research finds that that factor is significantly associated with drug use.
However, drug use during sex could also be due to anxiety about sexual intimacy, and make a brief reference to that.
Finally, I expect the authors to extend the practical implications, perhaps going on to suggest some possible protocols or therapeutic intervention strategies. For example. one could mention working with schema therapy by considering the issue of abandonment and rejection (https://doi.org/10.3390/ijerph21080971). Finally, directions for future research should also be extended: what are the prevalent methodological limitations that characterize the studies, and how future researchers can overcome these limitations. In addition, again in this area, it would be of interest to cosnigliare theoretical constructs for inclusion in future investigations.
Bibliographic suggestions are recommended and I would be happy to cite them. However, they are mandatory and will not affect the subsequent evaluation of the paper.
Author Response
Responses to Reviewer 2
Comments and Suggestions for Authors
I was very happy to read this scoping review. The method is appropriate when one wants to attempt an initial exploration within a still poorly reviewed literature. Important thoughts and insights for future research and clinical and prevention practice emerge from the authors' work. The introduction is pertinent (perhaps I would reduce the part on drug emergence, which is an established fact) and also include the concept of internalized homophobia as a factor that impacts the psychological health of young people (https://doi.org/10.1080/19419899.2018.1476905) and could be the consequence of rejection by peers, peers and the community at large.
Response: The authors would like to sincerely thank the reviewer for such positive remarks.
The introduction has now been adjusted to reflect on some of the comments including less content in relation to drug emergence and the addition of evidence on internal homophobia/ homonegativity. Great suggestion that add strength to the cause of this review.
Badenes-Ribera, L., Fabris, M. A., & Longobardi, C. (2018). The relationship between internalized homonegativity and body image concerns in sexual minority men: a meta-analysis. Psychology & Sexuality, 9(3), 251–268. https://doi.org/10.1080/19419899.2018.1476905
The authors argued the criteria for inclusion and exclusion well and also explained why they focused only on such a narrow time frame. Usually in scoping reviews, gray literature is also used, but because scoping reviews focused only on peer-reviwed articles are flourishing, I ask the authors to specify this fact.
Response: Good point by the review in relation to the use of gray literature. We have found sufficient reliable peer-review evidence to answer the research question and refrained from using any gray literature to provide uniformity of the evidence. We also excluded gray literature as it can be time-consuming and this was somewhat supported in modern framework.
I would avoid stating for each specific factor the sample size of the included research. In addition, I would prepare a table summarizing the factors identified and the studies that identified them, perhaps inserting with an X when a particular research finds that that factor is significantly associated with drug use.
Response: Great observation re the numbers of participants in each study/ category. We have now omitted that part from the paper. Thank you. Re: a table to summarise factors identified in each study, it’s a great idea and we will endeavor to do this in our future reviews. For now, we hope that the comprehensive table 1 would suffice. Great suggestion.
However, drug use during sex could also be due to anxiety about sexual intimacy, and make a brief reference to that.
Response: we did not understand where this could fit- is this in relation to the background/ intro of the determinants of drug use in this community?
Finally, I expect the authors to extend the practical implications, perhaps going on to suggest some possible protocols or therapeutic intervention strategies. For example. one could mention working with schema therapy by considering the issue of abandonment and rejection (https://doi.org/10.3390/ijerph21080971).
Response: Great suggestion by the reviewer. This evidence for implications is now added to the discussion and moving forward part.
Cardoso BLA, Lima AFA, Costa FRM, Loose C, Liu X, Fabris MA. Sociocultural Implications in the Development of Early Maladaptive Schemas in Adolescents Belonging to Sexual and Gender Minorities. International Journal of Environmental Research and Public Health. 2024; 21(8):971. https://doi.org/10.3390/ijerph21080971
Finally, directions for future research should also be extended: what are the prevalent methodological limitations that characterize the studies, and how future researchers can overcome these limitations. In addition, again in this area, it would be of interest to cosnigliare theoretical constructs for inclusion in future investigations.
Response:
Another value-adding remark by the reviewer. We have revised the discussion and future directions by commenting on the gap in research and suggestions for more work including systematic reviews to confirm the findings from this review.
Bibliographic suggestions are recommended and I would be happy to cite them. However, they are mandatory and will not affect the subsequent evaluation of the paper.
Response: thank you, we have incorporated 2 citations as suggested by the reviewer which we believe added value.
once again, thanks for the extensive review that has undoubtedly added value to our manuscript.
Reviewer 3 Report
Comments and Suggestions for Authors
The purpose of this manuscript was to evaluate the determinants of drug use among LGBTQIA+ adolescents. The focus of the study fits within the scope of the journal and is of relevance to its readership and provides important information for the development of interventions to combatting drug use among LGBTQIA+ adolescents. Below, I note specific areas to strengthen the manuscript and note concerns about. I wish the authors the best of luck as they continue to move this research forward.
Overall
1. The title uses the term drug use, but substance use and illicit drug use are used throughout. I would suggest selecting one term and using consistently throughout.
Introduction
2. Is the information in the first paragraph (lines 33-50) in reference to adolescents? If not, the authors may consider tailoring this information to adolescents as it currently reads as very broad.
3. Sentence beginning in line 56 needs citation(s).
4. The authors spend the majority of the paragraph beginning at line 56 talking about mental scars/mental illness, but then proceed to state that “A recent study showed a significant association between identity (sexual orientation or gender) related distress and drug use, especially among queer adolescents (Goldbach et al. 2014).” First, I am unsure about the focus on only mental health as a precursor to drug use and second, I am not sure how that supports the above quote. I think it would be important to discuss the various factors (or determinants) that may be related (including mental health and sexual identity) to drug use, particularly the ones (e.g., peer-peer interactions, homelessness) that are mentioned in the abstract and later on in the paper.
5. Relatedly, I am not sure how the last sentence (lines 71-73) is related to the paragraph.
6. Overall, the introduction is quite brief and could focus more one information that is pertinent specifically to LGBTQIA+ adolescents.
Methods
7. The authors should provide a citation in the first sentence for JBI methodology presented.
8. Can the authors please describe who conducted the article screening and their qualifications and/or experience in the content area?
Results
3.2.2. Peer-peer interactions
9. Can the authors please elaborate on the definition of peer-to-peer interactions? Interactions between individuals is very broad. Are they the same age? Any other similarities among these “peers”? In the context of adolescence, peers are typically considered classmates, friends, etc.
10. All the examples in this section are related to (sexual) violence. Unclear why peer-peer interactions is used as opposed to “Violence.”
3.2.3 Mental health
11. It does not seem as though the studies referenced in this section are longitudinal and that these inferences suggesting causality (mental health being a determinant of drug use) can be made.
Discussion
12. The Discussion seems the quite brief and somewhat repetitive of the information presented in the Results. Implications for real-world practice would be helpful and provide more substance to the discussion.
Author Response
Reviewer 3:
Comments and Suggestions for Authors
The purpose of this manuscript was to evaluate the determinants of drug use among LGBTQIA+ adolescents. The focus of the study fits within the scope of the journal and is of relevance to its readership and provides important information for the development of interventions to combatting drug use among LGBTQIA+ adolescents. Below, I note specific areas to strengthen the manuscript and note concerns about. I wish the authors the best of luck as they continue to move this research forward.
Overall
- The title uses the term drug use, but substance use and illicit drug use are used throughout. I would suggest selecting one term and using consistently throughout.
Response: We thank the review for this great point and we agree and have therefore adjusted the title to reflect substance use which is more appropriate term and consistent with the content.
Introduction
- Is the information in the first paragraph (lines 33-50) in reference to adolescents? If not, the authors may consider tailoring this information to adolescents as it currently reads as very broad.
Response: This introductory statement is designed to be a bold first statement highlighting the risk before delving into the focused topic in the younger population. We hope this is ok with the reviewer
- Sentence beginning in line 56 needs citation(s).
Response: Great point. We have corrected this by adding a reference to reflect this statement.
Hatzenbuehler, M. L., & Pachankis, J. E. (2016). Stigma and Minority Stress as Social Determinants of Health Among Lesbian, Gay, Bisexual, and Transgender Youth: Research Evidence and Clinical Implications. Pediatric clinics of North America, 63(6), 985–997. https://doi.org/10.1016/j.pcl.2016.07.003
- The authors spend the majority of the paragraph beginning at line 56 talking about mental scars/mental illness, but then proceed to state that “A recent study showed a significant association between identity (sexual orientation or gender) related distress and drug use, especially among queer adolescents (Goldbach et al. 2014).” First, I am unsure about the focus on only mental health as a precursor to drug use and second, I am not sure how that supports the above quote. I think it would be important to discuss the various factors (or determinants) that may be related (including mental health and sexual identity) to drug use, particularly the ones (e.g., peer-peer interactions, homelessness) that are mentioned in the abstract and later on in the paper.
Response: We have made modification to the section (see highlights) we hope it satisfies the reviewer’s comment.
- Relatedly, I am not sure how the last sentence (lines 71-73) is related to the paragraph.
Response: We have modified the statement to reflect our thoughts behind it. The idea of this line was to demonstrate that adolescence in general is a sensitive age where damage can be caused- see modified statement “During adolescence, the brain's cognitive functions are still in the process of developing, which makes all young people more vulnerable to experimentation and potential addiction. This period of heightened susceptibility is not unique to any one group, and adolescents identifying as LGBTQ+ experience the same neurodevelopmental challenges as their peers.”
- Overall, the introduction is quite brief and could focus more one information that is pertinent specifically to LGBTQIA+ adolescents.
Response: Good point. We made some modifications to reflect the reviewer’s wish and have added value for sure- check added citation too.
Rulison, Kelly, Megan E. Patrick, and Jennifer Maggs, 'Linking Peer Relationships to Substance Use Across Adolescence', in Robert A. Zucker, and Sandra A. Brown (eds), The Oxford Handbook of Adolescent Substance Abuse, Oxford Library of Psychology (2019; online edn, Oxford Academic, 10 Dec. 2015), https://doi.org/10.1093/oxfordhb/9780199735662.013.019, accessed 19 Oct. 2024.
Methods
- The authors should provide a citation in the first sentence for JBI methodology presented.
Response: Done – see newly done methodology section
- Can the authors please describe who conducted the article screening and their qualifications and/or experience in the content area?
Response: Great point. This has now been added to the paper.
EB (MBBS, MPH) and EA (B.Sc Pharm. PhD Pharm) conducted the database search and the screening for the studies while FH performed a final review of the process.
Results
3.2.2. Peer-peer interactions
- Can the authors please elaborate on the definition of peer-to-peer interactions? Interactions between individuals is very broad. Are they the same age? Any other similarities among these “peers”? In the context of adolescence, peers are typically considered classmates, friends, etc.
Response: Great suggestion. This section has been added to the discussion elaborating on peer interaction with a reference in fulfillment of the reviewer’s point:
Response: DISCUSSION section “Peers can greatly influence behaviors during this developmental stage. Research shows that adolescents often engage in substance use due to peer pressure, modeling behaviors, or seeking acceptance within their social groups. These interactions shape attitudes toward substances, normalizing or encouraging use among young people (Rulison, 2024)
Rulison, Kelly, Megan E. Patrick, and Jennifer Maggs, 'Linking Peer Relationships to Substance Use Across Adolescence', in Robert A. Zucker, and Sandra A. Brown (eds), The Oxford Handbook of Adolescent Substance Abuse, Oxford Library of Psychology (2019; online edn, Oxford Academic, 10 Dec. 2015), https://doi.org/10.1093/oxfordhb/9780199735662.013.019, accessed 19 Oct. 2024.
- All the examples in this section are related to (sexual) violence. Unclear why peer-peer interactions is used as opposed to “Violence.”
Response: Good question. We think that violent peer-peer interaction in adolescents of LGBTQIA+ tends to be associated with sexual behaviour and particularly with the involvement of drugs.
3.2.3 Mental health
- It does not seem as though the studies referenced in this section are longitudinal and that these inferences suggesting causality (mental health being a determinant of drug use) can be made.
Response: we have now added this as a limitation that some of these studies were cross-sectional in nature and causation between mental health and drug use may not have been completely evident. However, given the magnitude of the results and the agreement of multiple studies findings in this review, the link between mental health issues and drug use is articulated!
"The review highlights a consistent association between mental health issues and drug use across multiple studies, suggesting a potential link between these conditions."
"The observed associations across diverse studies indicate a relationship between mental health symptoms and substance use behaviors."
"However, the absence of longitudinal studies limits the ability to establish a causal relationship, making it unclear whether mental health issues precede drug use or vice versa."
Response: Agreed. Lack of longitudinal studies may impact the evidence and makes it impossible to ascertain causality- we have added this to the limitations of the study
Discussion
- The Discussion seems the quite brief and somewhat repetitive of the information presented in the Results. Implications for real-world practice would be helpful and provide more substance to the discussion.
Response: We have expanded on the discussion in fulfillment of the reviewer's comment. Please see new highlighted sections
Once again, we thank the reviewer for the extensive review which has added value to our manuscript. excellent work
Reviewer 4 Report
Comments and Suggestions for Authors
Review of ijerph-3090315 " Evaluating the determinants of drug use in LGBTQIA+ adolescents: a scoping review "
The present manuscript described a scoping review of the literature on illicit drugs use among adolescent LGBTQ+ individuals. The authors described their search strategy and identified “themes” of the literature they found. The manuscript deals with an important research topic. However, I do not think this manuscript is particularly impactful. The results do not meaningfully add to the minority stress literature. What follows are a list of my critiques of the manuscript.
1. There are several instances where the authors make statements/claims without including needed citations. For example, on lines 64-66 the authors state “To cope with mental health issues, stigma, or prejudice, many LGBTQIA+ people often turn to unhealthy coping mechanisms, so to lessen the daily pressure exerted on their mental health and wellbeing. One common mechanism explored by the LGBTQIA+ community is the use of illicit drugs.” Some LGBTQ+ people may use drugs as a coping mechanism but to suggest that this happens “often” is a bold claim to make without citing evidence.
2. The manuscript would benefit from editing to update language/terminology. For example, the authors should not be using the term “transexuals”.
3. The broadness of search strategy limits the usefulness of the findings. For example, the research cited comes from several different countries (e.g., the US, New Zealand, Australia, South Africa, Singapore, and Brazil) with very different drug laws. I think it is valid to draw broad conclusions about human behavior based on research from different countries, but it is unclear to me how this can inform public health interventions specifically for LGBTQ+ adolescents. Also, what drugs are illegal varies drastically across countries making it challenging to draw meaningful conclusions across samples,
4. The manuscript does not add to the Minority Stress literature. There have been multiple studies examining risk factors of substance use among LGBTQ+ young people (e.g., https://doi.org/10.1016%2Fj.copsyc.2019.05.002). I do not see how this manuscript meaningfully adds to the field.
Author Response
Reviewer 4:
Comments and Suggestions for Authors
Review of ijerph-3090315 " Evaluating the determinants of drug use in LGBTQIA+ adolescents: a scoping review "
The present manuscript described a scoping review of the literature on illicit drugs use among adolescent LGBTQ+ individuals. The authors described their search strategy and identified “themes” of the literature they found. The manuscript deals with an important research topic. However, I do not think this manuscript is particularly impactful. The results do not meaningfully add to the minority stress literature. What follows are a list of my critiques of the manuscript.
- There are several instances where the authors make statements/claims without including needed citations. For example, on lines 64-66 the authors state “To cope with mental health issues, stigma, or prejudice, many LGBTQIA+ people often turn to unhealthy coping mechanisms, so to lessen the daily pressure exerted on their mental health and wellbeing. One common mechanism explored by the LGBTQIA+ community is the use of illicit drugs.” Some LGBTQ+ people may use drugs as a coping mechanism but to suggest that this happens “often” is a bold claim to make without citing evidence.
Response: thank you for this important observation… we have now added Evidence or lessen the tone … https://www.psychologytoday.com/us/blog/queering-health/202306/minority-stress-and-lgbtqia-peoples-mental-health
- The manuscript would benefit from editing to update language/terminology. For example, the authors should not be using the term “transexuals”.
Response: This is another great observation by the reviewer… the term transsexual has been replaced by “transgender”
- The broadness of search strategy limits the usefulness of the findings. For example, the research cited comes from several different countries (e.g., the US, New Zealand, Australia, South Africa, Singapore, and Brazil) with very different drug laws. I think it is valid to draw broad conclusions about human behavior based on research from different countries, but it is unclear to me how this can inform public health interventions specifically for LGBTQ+ adolescents. Also, what drugs are illegal varies drastically across countries making it challenging to draw meaningful conclusions across samples,
Response: This is added to the limitations based on author’s remark.
- The manuscript does not add to the Minority Stress literature. There have been multiple studies examining risk factors of substance use among LGBTQ+ young people (e.g., https://doi.org/10.1016%2Fj.copsyc.2019.05.002). I do not see how this manuscript meaningfully adds to the field.
Response: Thank you for the comment. Our review covers longer-term literature than the above study that was confined to 2 years of literature 2016-2018. Also, the sole author of the above review ( E Mereish) acknowledged that more work is required to establish evidence among sub-groups, which is something our review had done and with literature analysis on a broader and more recent evidence. “concise and select review of the literature in PsycInfo and PubMed on substance use and misuse among sexual and gender minority youth (SGMY) published from 2016 to 2018
thanks once again for the added value to our manuscript.
Round 2
Reviewer 2 Report
Comments and Suggestions for Authors
Very well!
Thank you for your review! :)
Reviewer 3 Report
Comments and Suggestions for Authors
The authors did a commendable job in addressing the comments raised in the revision. I have no further comments.